# MedTric : A clinically applicable metric for evaluation of multi-label computational diagnostic systems

**Soumadeep Saha**[1,2]*, **Utpal Garain**[1], **Arijit Ukil**[2], **Arpan Pal**[2], **Sundeep Khandelwal**[2]

**1** Computer Vision and Pattern Recognition Unit, Indian Statistical Institute, Kolkata, West Bengal, India, **2** TCS Research, Tata Consultancy Services, Kolkata, West Bengal, India

\* soumadeep.saha_r@isical.ac.in

**Data Availability Statement:** The data underlying the results presented in the study are available from https://moody-challenge.physionet.org/2020/

## Abstract

When judging the quality of a computational system for a pathological screening task, several factors seem to be important, like sensitivity, specificity, accuracy, etc. With machine learning based approaches showing promise in the multi-label paradigm, they are being widely adopted to diagnostics and digital therapeutics. Metrics are usually borrowed from machine learning literature, and the current consensus is to report results on a diverse set of metrics. It is infeasible to compare efficacy of computational systems which have been evaluated on different sets of metrics. From a diagnostic utility standpoint, the current metrics themselves are far from perfect, often biased by prevalence of negative samples or other statistical factors and importantly, they are designed to evaluate general purpose machine learning tasks. In this paper we outline the various parameters that are important in constructing a clinical metric aligned with diagnostic practice, and demonstrate their incompatibility with existing metrics. We propose a new metric, MedTric that takes into account several factors that are of clinical importance. MedTric is built from the ground up keeping in mind the unique context of computational diagnostics and the principle of risk minimization, penalizing missed diagnosis more harshly than over-diagnosis. MedTric is a unified metric for medical or pathological screening system evaluation. We compare this metric against other widely used metrics and demonstrate how our system outperforms them in key areas of medical relevance.

## 1 Introduction

Machine learning techniques have shown great promise in computational diagnostics ([1, 2]) and have been applied to a wide set of diagnostic problems (e.g. [3, 4]), which are often "multi-label", i.e. where several diagnostic features might be detected from one data sample [5]. For instance, consider a blood sample which might be evaluated by a pathologist to detect presence of several pathogens, or a radiologist marking various anomalies in a CT scan. From an aggregated health-care cost perspective, the potential benefit from algorithmic screening can be massive (see Fig 1), provided we can find a suitable system. Therefore, comparing several competing computational diagnostic systems in accordance with clinical outcomes is paramount for deployment in clinical applications. This however continues to pose a challenge.

https://stanfordmlgroup.github.io/competitions/chexpert/ https://www.uco.es/kdis/mllresources/.

**Funding:** The author(s) received no specific funding for this work.

**Competing interests:** The authors have declared that no competing interests exist.

There is no consensus on a good choice of metric [3], and it is recommended that models are evaluated on several metrics [6]. Significant strides in multi-label diagnostics have to grapple with this issue ([3, 4, 7]). This scatter shot approach, however, raises further problems, as models evaluated on different sets of metrics cannot be compared [8]. Worse still, the choice of metric can serve to highlight key strengths of a model, and sweep weaknesses under the rug [8]. Different metrics do not agree on comparative performance of systems either [9], thus the choice of best diagnostic system can be dictated by choice of metric. Additionally, even if results are reported on several metrics, it is not necessarily informative enough from a clinical perspective. A large set of scores, measuring different aspects of performance does not help us answer the question "Which system is better for clinical application?". Since the metrics are borrowed from machine learning, where requirements are different, a higher score on a certain metric does not necessarily translate to better diagnostic performance, and vice versa. Thus, it is imperative to have a metric that can order computational diagnostic models based on desired clinical outcomes [10].

In clinical practice some facts are ubiquitous, and can be treated like axioms. For instance, a wrong diagnosis (ground truth and prediction have no overlap) is worse than a missed-diagnosis (prediction is a subset of ground truth) which is in turn worse than over-diagnosis (ground truth is a subset of prediction) up to a certain extent. The standard metrics used in a multi-label setting (Hamming Loss, subset accuracy, etc) does not reflect this.

There might also be a scenario where certain sets of diagnostics have similar treatment plans and outcomes [10], thus making certain types of missed diagnosis less deleterious. Additionally, if there are $k$ possible diagnoses, all $2^k$ might not be feasible or be logically sound (for instance Sinus Tachycardia and Sinus Bradycardia or hypo- and hypertension).

The principle of risk avoidance would dictate that, in a computational diagnostic system, sensitivity should be correlated to cost; with significant ailments having markedly higher sensitivity than minor issues. However, when this comes at a cost of specificity, it might lead to alarm fatigue. Thus, a general multi-label metric does not align with the highly context dependent clinical principles and practice when rating a system, and is unable to capture the critically important features that ought to be present in a diagnostic system.

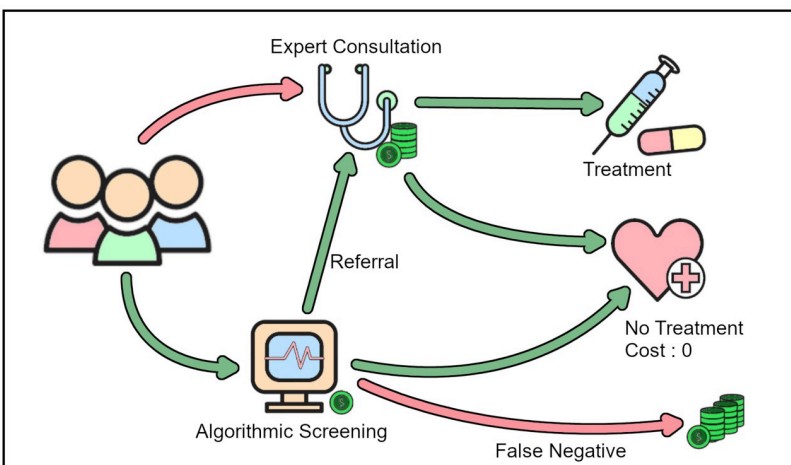

**Fig 1. Algorithmic screening cuts down on health-care costs.** Since algorithmic screening is orders of magnitude cheaper than expert intervention, well designed computational systems can make health-care accessible to a larger population.

Some metric in use today have other contentious elements [8]. Some are skewed by prevalence of negative samples [11], which is par for the course in diagnostic datasets, while some are biased towards high specificity [6].

Keeping, the clinical considerations in mind, and in consultation with experts from the domain we have outlined the qualities a clinically aligned metric should demonstrate.

- Missed diagnosis is more harmful than over-diagnosis.

- Wrong diagnosis is more harmful than over-diagnosis and missed diagnosis.

- Some diagnoses have more clinical significance.

- Some diagnoses are contradictory, and should be disqualifying.

- Quality of a diagnostic tool, should not depend on relative proportions of diseases present in the population (dataset distribution independence).

These criteria often come up in the context of cost-sensitive learning, where all misclassifications aren't equally weighted [12]. Our proposed framework extends this to a clinical context, and not only weighs disparate misclassifications differently, but also weighs the potential "cross-contaminations" in a clinically sound way.

In our comb-through of the literature, we did not come across a metric that satisfies all of the above-mentioned criteria. In the following section, we provide some basic definitions and descriptions of prevalent metrics. Then we illustrate their limitations and subsequently define our metric in keeping with the foundational principles, and demonstrate that its properties are in accordance with clinical demands. This is followed by a section on experimental design and results from comparison of several metrics in relevant clinical scenarios carried out on three public multi-label diagnostic datasets and a few concluding remarks.

## 2 Theoretical background

### 2.1 Definitions

**Definition** (Dataset). Given a set of diagnostic samples and their respective annotations $\mathcal{D} = \{(x_i, y_i) \mid i \in 1, \cdots, N\}$ where $x_i, y_i$ is the $i^{th}$ diagnostic sample and label respectively is called the **dataset** (See Table 1). Each $y_i$ is a set of diagnoses (drawn from a fixed set of possible diagnoses A = $\{a_1, a_2, \ldots a_P\}$), i.e. $y_i \in 2^A$.

**Definition** (Classifier).
$f_\theta : \mathcal{R} \rightarrow 2^A$   ($x_i \in \mathcal{R}$ is the space from which samples are drawn) is called a **classifier**. Given a diagnostic sample $x_i$ it attempts to recreate the corresponding label $y_i \approx \hat{z}_i = f_\theta(x_i)$ for some (potentially hidden) parameters $\theta$.

**Table 1. List of symbols.**

| Symbol | Meaning |
|---|---|
| A | Set of all possible diagnoses (finite) \|A\| = $P$ |
| $a_j \in$ A | $j$-th diagnosis in the diagnosis set. |
| $2^A$ | The set of all possible subsets of A. |
| $\mathcal{D}$ | The diagnostic dataset. $\|\mathcal{D}\| = N$ |
| $(x_i, y_i) \in \mathcal{D}$ | The $i$-th instance from the dataset with data $x_i \in \mathcal{R}$ and label $y_i \in 2^A$. |
| $f_\theta(.)$ | The diagnostic system. |
| $\hat{z}_i \in 2^A$ | The prediction set for $(x_i, y_i)$ given by $f_\theta$ |
| $\mathcal{M}(.,.)$ | A metric; maps $\{(\hat{z}_i, y_i)\}$ to a score $s \in \mathbb{R}$ |

More often than not, we have outputs from the classification scheme in the form of scores, which are correlated to probabilities of a certain diagnostic condition being present, i.e. $g_\theta : \mathcal{R} \to [0, 1]^P$ and some thresholding protocol, $t(x_i) \mapsto t_i \in [0, 1]^P$. A successful scheme should have

$$g_\theta(x_i)_j > g_\theta(x_i)_k \text{ if } a_j \in y_i \text{ and } a_k \notin y_i \tag{1}$$

We also define

$$z_{i(k)} = h_\theta(x_i)_{(k)} := \begin{cases} 1 \text{ if } g_\theta(x_i)_{(k)} > t_{i(k)} \\ 0, \text{ otherwise} \end{cases} \tag{2}$$

The prediction set $\hat{z}_i$, i.e. the set of all diagnostic conditions meeting the prediction threshold, is given by $\hat{z}_i = \{a_j | z_{ij} = 1, \ \forall j \in \{1, \ldots, P\}\}$. This process involving the function $g_\theta$ and the thresholding protocol together gives us a classifier. For the purpose of this paper, we consider only the output set $\hat{z}_i$ for evaluation, in order to have the most general treatment of various kinds of computational diagnostic systems, and the classification system described above involving the function $g_\theta$ and the thresholding protocol together is taken as a black box.

**Definition** (Wrong Diagnosis). A prediction $\hat{z}_i$ is said to be a wrong diagnosis, if $\hat{z}_i \cap y_i = \phi$, i.e. prediction and ground truth are disjoint.

**Definition** (Missed Diagnosis). A prediction $\hat{z}_i$ is said to be a missed diagnosis, if $\hat{z}_i \subsetneq y_i$, i.e. prediction is a proper subset of ground truth labels.

**Definition** (Over Diagnosis). A prediction $\hat{z}_i$ is said to be an over-diagnosis, if $y_i \subsetneq \hat{z}_i$, i.e. ground truth is a proper subset of predicted labels.

**Definition**. The elements of sets $\hat{z}_i - y_i, y_i - \hat{z}_i,$ and $y_i \cap \hat{z}_i$ are called extra predictions, missed predictions, and correct predictions respectively.

## 2.2 Metrics

To judge the quality of the classifier $f_\theta$ over the dataset $\mathcal{D}$, it is sufficient to analyze the set $\mathcal{P} = \{(\hat{z}_i, y_i) | \forall i, s.t. (x_i, y_i) \in \mathcal{D}\}$. The job of a metric, given such a set $\mathcal{P}$ is to provide a number, which is correlated to the performance of the classification system. We shall not be exploring metrics designed for the label ranking task [13] (coverage, AUC [8], etc), since they are not relevant in this context, instead we shall focus on *bipartition* based metrics in the ensuing discussion, which are designed for the task at hand.

In particular AUC, although useful in certain contexts, poses the additional challenge of requiring access to the implementation details of the classifier in question, which limits the class of diagnostic systems we can talk about. For instance, the inference algorithm might make the decision to detect a condition based on a complicated function of output probabilities instead of treating them each like an independent binary classification task. This makes an AUC computation disconnected from actual predictions. Thus, we restrict ourselves to implementation blind metrics, i.e. those that can be computed given just the output and target labels.

Bipartition metrics can be broadly divided into two categories—**label based** (see Table 2) and **example based** (see Table 3).

The example based metrics assign a score based on averages over certain functions of the actual and predicted label sets. Label based metrics on the other hand compute the prediction performance of each label in isolation and then compute averages over labels.

**Table 2. Label based metrics.**

| Metric | Definition |
|---|---|
| Macro-precision | $\frac{1}{P}\sum_{j=1}^{P} \frac{tp_j}{tp_j + fp_j}$ |
| Macro-recall | $\frac{1}{P}\sum_{j=1}^{P} \frac{tp_j}{tp_j + fn_j}$ |
| Macro-F1-score | $\frac{1}{P}\sum_{j=1}^{P} F1_j, F1_j = \frac{2 p_j r_j}{p_j + r_j}$ |
| Micro-precision | $\frac{\sum_{j=1}^{P} tp_j}{\sum_{j=1}^{P} tp_j + \sum_{j=1}^{P} fp_j}$ |
| Micro-recall | $\frac{\sum_{j=1}^{P} tp_j}{\sum_{j=1}^{P} tp_j + \sum_{j=1}^{P} fn_j}$ |
| Micro-F1-score | $\frac{2 \cdot micro-precision \cdot micro-recall}{micro-precision + micro-recall}$ |

Certain other binary metrics have been proposed in a clinical diagnostic context, like threat score [14] or Mathews Correlation Coefficient [15], and we can define macro/micro averages or example based metrics based on these, however due to their limited usage in a multi-label context, we omit these (Giraldo-Forero, et al. [6] noted that that $F_1$ and MCC are closely related). Their definitions suggest that their behavior in key aspects follows the other metrics discussed in the following section.

**2.2.1 Label based metrics.** Label based metrics in use today [13] take the form of micro or macro averages of binary classification metrics (see Table 2), such as precision, recall and $F_1$ (or the general $F_\beta$) to provide summary information of performance across several categories. Specificity is unsuited in the clinical domain, due to the class imbalance usually present in diagnostic datasets, where negative examples are plentiful [11].

A **macro averaged** measure is computed by first independently computing the binary metric for each class and then averaging over them. A **micro average** on the other hand will aggregate the statistics across classes and compute the final metric. However, both of these approaches have their own drawbacks.

The micro average favors classifiers with stronger performance on predominant classes whereas the macro average favors classifiers suited to detecting rarely occurring classes. In a clinical setting where it is very common for certain presentations to be very rare, the micro average measures are less meaningful, as it is the rare diseases that are often of most concern and would benefit greatly from intervention.

**Table 3. Example based metrics.**

| Metric | Definition |
|---|---|
| Hamming Loss | $\frac{1}{N}\sum_{i=1}^{N} \frac{1}{P}|\hat{z}_i \Delta y_i|$ |
| Accuracy | $\frac{1}{N}\sum_{i=1}^{N} \frac{|\hat{z}_i \cap y_i|}{|\hat{z}_i \cup y_i|}$ |
| Precision | $\frac{1}{N}\sum_{i=1}^{N} \frac{|\hat{z}_i \cap y_i|}{|y_i|}$ |
| Recall | $\frac{1}{N}\sum_{i=1}^{N} \frac{|\hat{z}_i \cap y_i|}{|\hat{z}_i|}$ |
| F1-score | $\frac{1}{N}\sum_{i=1}^{N} \frac{2 \times |\hat{z}_i \cap y_i|}{|\hat{z}_i| + |y_i|}$ |
| Subset Accuracy | $\frac{1}{N}\sum_{i=1}^{N} I(\hat{z}_i = y_i)$ |
| Challenge Metric [10] | $a_{jk} = \sum_{i=1}^{N} \frac{I(a_j \in \hat{z}_i \text{ and } a_k \in y_i)}{|\hat{z}_i \cup y_i|}$ |
| | $s_{unnorm} = \sum_{k=1}^{P} \sum_{j=1}^{P} a_{jk} w_{jk}$ |
| | $CM = \frac{s_{unnorm} - s_{inactive}}{s_{perfect} - s_{inactive}}$ |

This however falls apart if the majority class is of greatest concern, as a macro averaged metric would paint an undeserved optimistic picture of the diagnostic system performance.

**2.2.2 Example based metrics.** Example based metrics (see Table 3) in use [16] are specifically designed to pick out certain key features of a multi-label classifier. It is in general inadequate to compute just one or two metrics [3], as they each have individual properties which provide beneficial cues.

One notable recent work by Alday, et al. [10], set out to design a metric that takes clinical outcomes into account in a multi-label diagnostic setting. The metric was designed to evaluate several computational models, built to pick out a subset of diagnostic features from 12 lead ECG signals and 27 potential diagnostic classes many of which might be simultaneously present. In this metric, we first define the multi-class confusion matrix $A = [a_{jk}]$ as

$$a_{jk} = \sum_{i=1}^{N} a_{ijk} \tag{3}$$

where

$$a_{ijk} = \begin{cases} \dfrac{1}{|\{\hat{z}_i \cup y_i\}|}, & \text{if } c_k \in \hat{z}_i \text{ and } c_j \in y_i \\ 0, & \text{otherwise.} \end{cases} \tag{4}$$

Next we compute $t(Y, Z) = \sum_k \sum_j w_{jk} a_{jk}$ where $w_{jk}$ is the weight matrix giving partial rewards to incorrect guesses. $w_{jj} = 1$ and in general $0 < w_{jk} \leq 1$. The final score is given as

$$CM = \frac{t(Y, Z) - t(Y, X_{\{NSR\}})}{t(Y, Y) - t(Y, X_{\{NSR\}})} \tag{5}$$

Where $X_{\{NSR\}}$ is a prediction set where all predictions are the normal class $\{NSR\}$. This metric which is a weighted version of accuracy [17], is limited to be used on the PhysioNet 2020/21 dataset [10], however with additional domain knowledge inputs, can be used in different contexts.

All of these metrics however, do not adequately take clinical aspects into account, for example the fact that over-diagnosis is less harmful than missed diagnosis, or the criticality of the diagnosis. In the following section we shall go through a thorough analysis of the existing metrics from a clinical perspective.

## 3 Are ML metrics clinically applicable?

From the preceding discussion it is clear, that a collection of metrics is inadequate when it comes to making deployment decisions, and we need one metric with relevant characteristics to compare different computational models. In the ensuing discussion we will see that the metrics borrowed from machine learning aren't cognizant of clinical requirements. Our yardstick for determining the clinical relevance of metrics will be based on the criteria set in the introductory section. In particular, we will check whether wrong diagnosis (WD) is more heavily penalized than missed diagnosis (MD) which in turn is penalized worse than over-diagnosis (OD), while the perfect diagnosis (PD) scores best, i.e.

$$\text{score}_{WD} < \text{score}_{MD} < \text{score}_{OD} < \text{score}_{PD} \tag{ClinicalOrder}$$

## 3.1 Label based metrics

It is generally accepted that example based metrics are better suited for the multi-label evaluation task [6]. However, for the sake of thoroughness, we will analyze some label based metrics in widespread use.

In the ensuing discussion we shall consider four classifiers, and their corresponding output sets $\mathcal{P}_O, \mathcal{P}_M, \mathcal{P}_W,$ and $\mathcal{P}$, which only have over, missed, wrong, and perfect diagnoses respectively (e.g. in $\mathcal{P}_O$ we have $y_i \subsetneq \hat{z}_i, \forall(\hat{z}_i, y_i) \in \mathcal{P}_O$).

- **Macro precision, macro recall and macro $F_1$**

  Macro precision and macro recall cannot be used in isolation, as we are free to change one at the expense of the other. However, Macro $F_1$ which is a macro average of the harmonic means of precision and recall is a serviceable metric. Macro $F_1$ is defined as

$$MacroF_1(\mathcal{P}) = \frac{1}{P}\sum_{j=1}^{P}F_{1(j)}(\mathcal{P}) \tag{6}$$

$$F_{1(j)}(\mathcal{P}) = \frac{2 \cdot p_j(\mathcal{P}) \cdot r_j(\mathcal{P})}{p_j(\mathcal{P}) + r_j(\mathcal{P})} \tag{7}$$

  Where $p_j$, $r_j$ is precision and recall for the $j^{th}$ class respectively. Consider the case where $r_j(\mathcal{P}_M) \geq p_j(\mathcal{P}_O)$ (Note, $p_j(\mathcal{P}_M) = r_j(\mathcal{P}_O) = 1$). Then we have,

$$\frac{2 \cdot r_j(\mathcal{P}_M)}{1 + r_j(\mathcal{P}_M)} \geq \frac{2 \cdot p_j(\mathcal{P}_O)}{1 + p_j(\mathcal{P}_O)} \tag{8}$$

$$\frac{2 \cdot p_j(\mathcal{P}_M) \cdot r_j(\mathcal{P}_M)}{p_j(\mathcal{P}_M) + r_j(\mathcal{P}_M)} \geq \frac{2 \cdot p_j(\mathcal{P}_O) \cdot r_j(\mathcal{P}_O)}{p_j(\mathcal{P}_O) + r_j(\mathcal{P}_O)} \tag{9}$$

$$F_{1(j)}(\mathcal{P}_M) \geq F_{1(j)}(\mathcal{P}_O) \tag{10}$$

  If this holds for all $j$, we have the exact opposite inequality as desired, and even if it is only true for some $j$ no guarantees can be made that a system that always misses diagnoses is worse than one that always over-diagnosed.

- **Micro precision, micro recall and micro $F_1$**

  Similar to their macro counterparts, micro precision and recall cannot be used in isolation, but micro $F_1$ can be used independently to evaluate the quality of a computational diagnostic system. It is defined as

$$MicroF_1(\mathcal{P}) = \frac{2 \cdot \text{micro} - \text{precision} \cdot \text{micro} - \text{recall}}{\text{micro} - \text{precision} + \text{micro} - \text{recall}} \tag{11}$$

  Where,

$$MicroPrecision(\mathcal{P}) \quad = \frac{\sum_{j=1}^{P} tp_j}{\sum_{j=1}^{P} tp_j + \sum_{j=1}^{P} fp_j} \tag{12}$$

$$MicroRecall(\mathcal{P}) \quad = \frac{\sum_{j=1}^{P} tp_j}{\sum_{j=1}^{P} tp_j + \sum_{j=1}^{P} fn_j} \tag{13}$$

We know $fp_j = 0$ in $\mathcal{P}_M$ and $fn_j = 0$ in $\mathcal{P}_O$. So,

$$MicroPrecision(\mathcal{P}_M) = 1 \tag{14}$$

$$\Rightarrow MicroF_1(\mathcal{P}_M) = \frac{2 \cdot MicroRecall(\mathcal{P}_M)}{1 + MicroRecall(\mathcal{P}_M)} \tag{15}$$

And, similarly,

$$MicroRecall(\mathcal{P}_O) = 1 \tag{16}$$

$$\Rightarrow MicroF_1(\mathcal{P}_O) = \frac{2 \cdot MicroPrecision(\mathcal{P}_O)}{1 + MicroPrecision(\mathcal{P}_O)} \tag{17}$$

So we have $MicroF_1(\mathcal{P}_M) \geq MicroF_1(\mathcal{P}_O)$ whenever,
$MicroRecall(\mathcal{P}_M) \geq MicroPrecision(\mathcal{P}_O)$. This means, if two diagnostic systems have the same number of true positives and one has higher number of false positives than the other has false negatives $MicroF_1(\mathcal{P}_M) \geq MicroF_1(\mathcal{P}_O)$. This is the opposite of the desired ordering in clinical practice as false negatives are generally more deleterious.

## 3.2 Example based metrics

In the ensuing discussion we consider predictions $m_i$, $o_i$, and $w_i$ which are missed, over, and wrong diagnosis respectively for the ground truth label $y_i$ and check if Clinical Order holds. (Note: $m_i \subsetneq y_i$, $y_i \subsetneq o_i$, and $y_i \cap w_i = \phi$).

- **Hamming Loss** is defined as

$$hloss(\mathcal{P}) = \frac{1}{N} \sum_{i=1}^{N} \frac{1}{P} |\hat{z}_i \Delta y_i| \tag{18}$$

So, from the definition it follows,

$$hloss(\{(m_i, y_i)\}) = hloss(\{(o_i, y_i)\}) \text{ whenever } |y_i - m_i| = |o_i - y_i| \tag{19}$$

So, missing $k$ diagnoses is penalized just as harshly as producing $k$ over-diagnoses. Since classifiers are tuned to target certain metrics, it must be noted that Hamming loss is usually not optimal for sensitive systems [6].

- **Accuracy** is widely known to be an unreliable measure in a clinical context, where imbalanced datasets are the norm. [11] It is defined as

$$accuracy(\mathcal{P}) = \frac{1}{N} \sum_{i=1}^{N} \frac{|\hat{z}_i \cap y_i|}{|\hat{z}_i \cup y_i|} \tag{20}$$

So, if

$$|m_i| \cdot |o_i| \geq |y_i|^2 \tag{21}$$

$$\Rightarrow accuracy(\{(m_i, y_i)\}) \geq accuracy(\{o_i, y_i\}) \tag{22}$$

Thus Clinical Order doesn't hold in general. As an example consider $|y_i| = k \geq 2$, $|m_i| = k - 1$, $|o_i| = k + 2$, then $accuracy(\{(m_i, y_i)\}) = \frac{k-1}{k} \geq \frac{k}{k+2} = accuracy(\{(o_i, y_i)\})$ [11].

- **Subset Accuracy** is the strictest metric, and is defined as

$$SAccuracy(\mathcal{P}) = \frac{1}{N} \sum_{i=1}^{N} I(\hat{z}_i = y_i) \tag{23}$$

So we have,

$$SAccuracy(\{(m_i, y_i)\}) = SAccuracy(\{(w_i, y_i)\}) = SAccuracy(\{(o_i, y_i)\}) = 0 \tag{24}$$

which violates Clinical Order.

- **$F_1$ score** is defined as

$$F_1(\mathcal{P}) = \frac{1}{N} \sum_{i=1}^{N} \frac{2 \times |\hat{z}_i \cap y_i|}{|\hat{z}_i| + |y_i|} \tag{25}$$

Suppose, $|y_i| = k$, $|m_i| = k - 1$ (one diagnosis missed) and $|o_i| = k + r$ ($r$ extra predictions).

$$F_1(\{(m_i, y_i)\}) \geq F_1(\{(o_i, y_i)\}) \text{ whenever } r \geq \left\lceil \frac{k}{k-1} \right\rceil \tag{26}$$

So, Clinical Order doesn't hold in general. As in the case of label based metrics, example based precision and recall aren't meaningful in isolation, and aren't discussed here.

- **PhysioNet 2020/21 Challenge Metric** Challenge Metric as defined in Eqs 4 and 5. Since, $w_{jk}$ is integral to the metric, it is limited for use on the PhysioNet 2020/21 Dataset, which is a multi-label 12 lead ECG dataset with 27 cardio-vascular diagnostic classes. Without the weight matrix (i.e. $w = I_{n \times n}$) this is the same as accuracy, and inherits all its problems. Even on the PhysioNet 2020/21 dataset it does not guarantee satisfaction of the inequality (Clinical Order). Of note are the issues introduced by their normalization scheme (as defined in 5). Consider the scenario where the ground truth label contains $y_i = \{NSR, a_j, a_k\}$ (NSR is the normal class), and we predict $\hat{z}_1 = \{NSR\}$, and $\hat{z}_2 = \{a_j\}$.

$$t(y, \hat{z}_1) = t(y, \hat{z}_{\{NSR\}}) \quad = \frac{1 + w_{j,NSR} + w_{k,NSR}}{3}$$

$$t(y, \hat{z}_2) \quad = \frac{1 + w_{NSR,j} + w_{k,j}}{3}$$

$$CM(y, \hat{z}_1) \quad = 0$$

$$CM(y, \hat{z}_2) \quad = \frac{1 + w_{NSR,j} + w_{k,j} - 1 - w_{j,NSR} - w_{k,NSR}}{3(t(y,y) - t(y, \hat{z}_1))} \tag{27}$$

$$CM(y, \hat{z}_2) \quad = \frac{w_{k,j} - w_{k,NSR}}{3(t(y,y) - t(y, \hat{z}_1))}$$

$$\Rightarrow CM(y, \hat{z}_2) \quad < 0, \text{ for some choice of } a_j$$

$$(\text{As } w_{jk} \text{ is symmetric .})$$

Therefore, CM discourages detection of cardiovascular conditions, in favor of detecting the normal class, which is contrary to clinical expectations.

# 4 Our proposal : MedTric

In the last section we demonstrated that most of the commonly used metrics aren't aligned with clinical practice. In this section we will propose a new metric which performs in accordance with the criterion laid out, and inculcates clinically desirable properties.

## 4.1 Definition

Given $\mathcal{P}$, consider an instance prediction and label $\hat{z}_i, y_i$. There are three sets of interest, $\hat{z}_i \cap y_i, y_i - \hat{z}_i, \hat{z}_i - y_i$ corresponding to correct predictions, missed predictions and extra predictions (see Fig 2). Although $y_i - \hat{z}_i, \hat{z}_i - y_i$ both consist of errors, the former generally has worse clinical outcomes.

Since, each category poses a unique clinical scenario, we score them as follows

$$a_{i(j)} = \begin{cases} \dfrac{s_j}{n_j} & \text{if } c_j \in \hat{z}_i \cap y_i \\[2mm] \dfrac{-s_j}{n_j} & \text{if } c_j \in y_i - \hat{z}_i \\[2mm] \dfrac{s_j}{n^*} \left[ \dfrac{1}{|y_i|} \left( \sum_{c_k \in y_i} w_{jk} \right) - 1 \right] & \text{if } c_j \in \hat{z}_i - y_i \\[2mm] 0 & \text{otherwise} \end{cases} \tag{28}$$

$n_i$ is the number of occurrences of diagnostic condition $i$ in the dataset $\mathcal{P}$, this ensures that prevalence of diagnostic conditions doesn't affect the final scores. $n^*$ is defined as follows

$$n^* = \max \{ n_j | \forall c_j \in y_i \} \tag{29}$$

$\{s_j | \forall j \in \{1 \ldots P\}\}$ are significance weights. This reflects the fact that all diagnostic conditions might not be equally relevant, and classes which are critical have a higher value of $s_i$, so their contribution to the final score is larger. They can all be set to 1 if their relative importance is the same.

$w_{jk}$ measures similarity of diagnostic conditions (as in Alday et al. [10]). This gives partial rewards to over diagnosis which are of similar nature in outcomes or treatment. If such a matrix is unavailable or not required, $w_{jk}$ can be set to a constant in $(0, 1)$ $(j \neq k, w_{jj} = 1 \; \forall j)$.

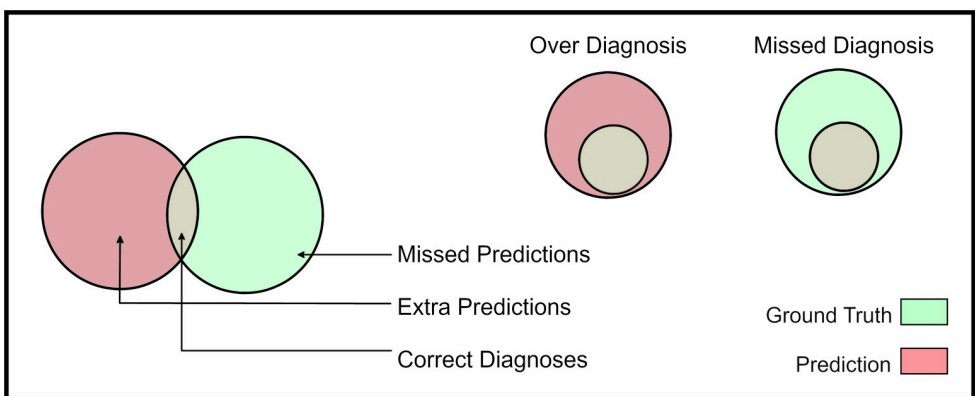

**Fig 2. The partitions of interest for clinical evaluation.**

If, for a given dataset having $P$ diseases, all $2^P$ diagnoses are not possible, and contradictory pairs exist (hypo- and hypertension for instance) we can introduce an additional contradiction penalty term and a contradiction matrix $C_{jk}$, such that $C_{jk} = 1$ if condition $c_j$ and $c_k$ can't occur together.

$$
b_{i(j)} = \begin{cases} \dfrac{-1}{n_j} \sum_{\forall k \text{ s.t. } c_k \in \hat{z}_i} s_k \cdot C_{jk} \text{ if } c_j \in \hat{z}_i \\ 0, \text{ otherwise} \end{cases}
\tag{30}
$$

Then we can compute the score for the $i^{th}$ instance, as follows

$$
t_i = \sum_{j=1}^{P} a_{i(j)} + b_{i(j)}
\tag{31}
$$

Finally, we sum the scores over all instances in the dataset, and normalize. Consider $Y, Z = \{y_i | \forall i \in \{1, \ldots, N\}\}, \{\hat{z}_i | \forall i \in \{1, \ldots, N\}\}$, we have $t(Y, Z)$ defined as follows.

$$
t(Y, Z) = \sum_{i=1}^{N} t_i
\tag{32}
$$

$$
M_{med} = \frac{t(Y, Z) - t(Y, \Phi)}{t(Y, Y) - t(Y, \Phi)}
\tag{33}
$$

Here $\Phi$ represents the null prediction set, i.e. $\hat{z}_i = \phi, \forall i \in \{1 \ldots N\}$. This normalization ensures that a perfect classifier gets a maximum possible score of 1 and an inactive one, that predicts nothing gets a score of 0.

## 4.2 Missed diagnosis vs wrong diagnosis vs over-diagnosis

**Claim**. MedTric always penalizes missed predictions more severely than extra predictions.
   *Proof*. Since, we have the following inequalities;

$$
0 < w_{jk} < 1 \quad \forall w_{jk}, j \neq k
\tag{34}
$$

$$
0 < \frac{1}{|y_i|} \sum_{c_k \in y_i} w_{jk} < 1
\tag{35}
$$

$$
-\frac{s_j}{n_j} < \frac{s_j}{n^*} \left[ \frac{1}{|y_i|} \left( \sum_{c_k \in y_i} w_{jk} \right) - 1 \right] < 0
\tag{36}
$$

$$
(n^* \geq n_j \forall j \in \{1 \ldots P\} \text{ by definition})
\tag{37}
$$

missed predictions always have heavier penalties than extra predictions.
   This does not demonstrate that MedTric follows Clinical Order, and since such a demonstration would be dependent on the exact clinical requirements and details about the dataset, we resort to empirical means in order to validate that Clinical Order is maintained by MedTric. However, MedTric does have desirable behavior in most cases of practical interest. Consider (as in Section 3.1) 4 classifiers and their output sets $\mathcal{P}_O, \mathcal{P}_M, \mathcal{P}_W$, and $\mathcal{P}$ corresponding to over, missed, wrong and perfect diagnosis respectively, and a specific diagnostic condition $a_k$.

In $\mathcal{P}_M$ since only missed diagnoses are allowed, we have $fp_k = 0, \hat{z}_i - y_i = \phi$ and the assigned score is given by $\frac{s_k}{n_k}(tp_k - fn_k)$ where $tp_k, fp_k, fn_k$ are the number of true positives, false positives and false negatives respectively for the condition $a_k$ in $\mathcal{P}_M$.

Similarly, in $\mathcal{P}_O$ since only over-diagnoses are allowed, we have $fn'_k = 0, y_i - \hat{z}_i = \phi, tp'_k = n_k$, and the assigned score is given by

$$
s_k + \sum_{a_k \notin y_i} \frac{s_k}{n^*} \left[ \frac{1}{|y_i|} \left( \sum_{c_j \in y_i} w_{kj} \right) - 1 \right]
$$

$$
\geq s_k + \sum_{a_k \notin y_i} \frac{s_k}{n^*} \left[ \frac{1}{|y_i|} \left( \sum_{c_j \in y_i} w_k^* \right) - 1 \right]
$$

$$
= s_k + \sum_{a_k \notin y_i} \frac{s_k}{n^*} \left[ (w_k^*) - 1 \right] \tag{38}
$$

$$
= s_k + fp'_k \frac{s_k}{n^*} (w_k^* - 1)
$$

$$
\text{Where } w_k^* = \min \left( w_{kj} \ \forall j \in \{1, \ldots, P\} \right)
$$

Where, $tp'_k, fp'_k, fn'_k$ are the number of true positives, false positives and false negatives respectively for the condition $a_k$ in $\mathcal{P}_O$. Consider,

$$
\xi_k = \overbrace{\frac{s_k}{n_k}(tp_k - fn_k)}^{\text{Missed diagnosis score}} \overbrace{- s_k - fp'_k \frac{s_k}{n^*}(w_k^* - 1)}^{\text{Over-diagnosis score}}
$$

$$
= \frac{s_k}{n_k} \left[ tp_k - fn_k - n_k - fp'_k \frac{n_k}{n^*}(w_k^* - 1) \right] \tag{39}
$$

$$
\text{as, } tp_k + fn_k = n_k
$$

$$
= \frac{s_k}{n_k} \left[ fp'_k \frac{n_k}{n^*}(1 - w_k^*) - 2 \cdot fn_k \right]
$$

Now, if $\xi_k < 0 \ \forall k \in \{1, \ldots, P\}$, MedTric follows Clinical Order. Even conservatively, since we have $\frac{n_k}{n^*} \leq 1$ and $0 < 1 - w_k^* < 1$ by definition, $\xi_k < 0$ holds whenever the number of false positives of each condition does not exceed twice the number of false negatives.

If a broader region of operation is required, $w_k^*$ can be adjusted accordingly, e.g. if $w_k^* = \frac{1}{3} \ \forall k$ MedTric follows Clinical Order whenever the number of false positives of each condition does not exceed thrice the number of false negatives. In more realistic scenarios however, where prevalence is imbalanced the region where Clinical Order holds is much broader. For example, if a certain diagnostic condition is a tenth as likely as the most frequent one, we have $\xi_k < 0$ whenever the number of false positives for the condition is less than 20 times the number of false negatives for that same condition.

For $\mathcal{P}_W$, we have $tp_k = 0, fn_k = n_k, \hat{z}_i \cap y_i = \phi$ and the score corresponding to $a_k$ is given by $-s_k - fp'_k \frac{s_k}{n^*}(1 - w_k^*) \leq -s_k$, which is the lowest possible missed diagnosis score.

Thus depending on the clinical context and its associated tolerance for missed diagnosis vs over-diagnosis, we can choose the values of $w_{jk}$ such that MedTric is guaranteed to follow Clinical Order (see example in Table 4).

**Table 4. Example of scoring for missed, over and wrong diagnoses.** O, M, W, P stands for over, missed, wrong and perfect diagnoses respectively, the following subscript number represents the quantity, e.g. $O_1$ means one over-diagnosis. *MedTric* sorts them in the desired clinical order (labels are drawn from PhysioNet dataset).

| Ground Truth | Prediction | Type | Score |
|---|---|---|---|
| CRBBB, AF, QAb | LAD, STach, TInv | $W_3$ | -0.230 |
| CRBBB, AF, QAb | LAD, STach | $W_2$ | -0.159 |
| CRBBB, AF, QAb | LAD | $W_1$ | -0.081 |
| CRBBB, AF, QAb | $\phi$ | $M_3$ | 0.0 |
| CRBBB, AF, QAb | CRBBB | $M_2$ | 0.25 |
| CRBBB, AF, QAb | CRBBB, AF | $M_1$ | 0.75 |
| CRBBB, AF, QAb | CRBBB, AF, QAb, LAD, NSIVCB | $O_2$ | 0.756 |
| CRBBB, AF, QAb | CRBBB, AF, QAb, LAD | $O_1$ | 0.918 |
| CRBBB, AF, QAb | CRBBB, AF, QAb | $P$ | 1 |

## 4.3 Clinical importance or dataset artifacts?

If a computational system is X% accurate in one diagnostic class and Y% in another, the scores of some metrics may change just by virtue of change of the proportion of each of these classes present. Micro averaged label based metrics and example based metrics are susceptible to this. Model performance measurement can be obfuscated by artifacts of demographics, especially since class imbalance is a prevalent problem in diagnostic datasets [18].

Since this is undesirable, we divide each score contribution by the corresponding class frequency (see Eqs 28 and 30), thus the final score is dataset proportion independent and is a reflection of the raw per-instance accuracy (see Table 5).

Our proposal fits neatly into the framework of cost sensitive learning. We saw in the preceding section, that our metric penalizes false negatives (missed diagnosis) more severely than false positives (over diagnosis). Additionally, we have disentangled prevalence of diagnostic conditions from performance measures as rarity is not essentially correlated to severity.

However, independently of prevalence, diagnostic datasets often have a notion of criticality, which is not captured in most machine-learning metrics. This notion of criticality requires another layer of cost based decision making. The significance weights $s_j$ (Eqs 28 and 30) ensures harsher penalties for classifiers which perform poorly on critical classes. These values are normalized, so it can be thought of as *the portion of the final score contributed to by a particular diagnostic class*.

With the introduction of $w_{jk}$ and $C_{jk}$ we also capture interactions between different diagnostic classes in a domain aware way. For example, if two diagnostic conditions share prognosis or treatment plans, we might weigh misclassification of one as the other less severely [10].

**Table 5. Example illustrating dataset prevalence independence.** Here in the two cases shown above the underlying classification quality is the same, conditions A and B are detected 100% of the time and condition X is detected 50% of the time, only the prevalence in the dataset has changed (in Case 1—{X, A} occurs 10% of the time and in Case 2—90% of the time). However unlike other metrics (e.g. F1 score), this doesn't change the MedTric score thus demonstrating dataset prevalence invariance.

| | Prediction | X, A | A | A,B | Total | F1 | MedTric |
|---|---|---|---|---|---|---|---|
| Case 1 | GT: X, A | 50 | 50 | 0 | 100 | 0.983 | |
| | GT: A, B | 0 | 0 | 900 | 900 | | 0.833 |
| | Prediction | X, A | A | A,B | Total | F1 | MedTric |
| Case 2 | GT: X, A | 450 | 450 | 0 | 900 | 0.850 | |
| | GT: A, B | 0 | 0 | 100 | 100 | | 0.833 |

Note that this can be phrased as a cost sensitive learning problem with a cost matrix $M \in \mathbb{R}^{2^P \times 2^P}$, such that every $\alpha \in 2^A$ being misclassified as $\beta \in 2^A$ has an associated (possibly distinct) cost.

## 5 Experiments

In the preceding section we demonstrated that our metric guarantees that Clinical Order is satisfied (i.e. scoring follows monotonic order given by clinical severity) under certain conditions. We also claimed that MedTric maintains this property in most cases of practical interest. Our analysis also indicates that other applicable metrics fail to follow Clinical Order, often in very commonplace scenarios. However, for a fair comparison, in this section, we will check each metric against a common set of relevant diagnostic scenarios to assert their suitability in a clinical context.

Metric scores are often dictated by the occurrence frequency of the various classes in the evaluation dataset, which might hide performance weaknesses in particular classes owing to their rarity. However, our metric by design, guarantees invariance of scores with change in prevalence of diagnostic conditions. In particular, we want to see how frequently are these conditions violated (if at all) by the various metrics in question. Since computational diagnostic systems, especially machine learning based methods, are tuned to certain metrics, it follows that if the metrics are inconsistent with clinical practice, models will follow suit.

In order to measure these we use three publicly available multi-label diagnostic datasets, from different diagnostic disciplines and modalities. The first is the PhysioNet Computation in Cardiology challenge 2020/21 dataset [10], where 27 cardiovascular conditions must be detected from 12-lead ECGs. The second is CheXpert [19]—a large chest radiograph dataset labeled with 14 classes of findings from frontal and lateral X-rays, and finally we used a multi-label free text classification dataset [20] labeled with 45 ICD-9 codes. Further details of the datasets can be found in S2 Appendix.

### 5.1 Monotonicity

In order to check violations of monotonicity we first sample a data point $(x_i, y_i)$ from the dataset $\mathcal{D}$ in question. Following this, we generate $\Gamma$ candidate predictions $\hat{z}_{i\gamma}$ such that

$$f_{\text{random}}(y_i) = \hat{z}_{i\gamma} = \{a_m | P(a_m \in \hat{z}_{i\gamma} | a_m \in y_i) \quad = p, \text{ and,}$$
$$P(a_n \in \hat{z}_{i\gamma} | a_n \notin y_i) \quad = 1 - q\} \tag{40}$$

This simple model $f_{\text{random}}$ emulates a classifier that has a sensitivity of $p$ and specificity of $q$ in each class. Next we group the predictions into several buckets, each with a particular type of diagnosis (wrong, missed, over or perfect) and the degree (count) of the same.

$$\overbrace{\hat{z}_{i(0)}, \hat{z}_{i(1)}, \ldots, \hat{z}_{i(c_1)}}^{t_w \text{ wrong}}, \ldots, \overbrace{\hat{z}_{i(c_2)}}^{t_w-1 \text{ wrong}}, \ldots, \overbrace{\ldots}^{\ldots}, \ldots, \overbrace{\hat{z}_{i(c_3)}, \ldots, \hat{z}_{i(c_4)}}^{1 \text{ wrong}},$$

$$\overbrace{\hat{z}_{i(c_4+1)}, \ldots, \hat{z}_{i(c_5)}}^{t_m \text{ missed}}, \ldots, \overbrace{\hat{z}_{i(c_6)}}^{t_m-1 \text{ missed}}, \ldots, \overbrace{\ldots}^{\ldots}, \ldots, \overbrace{\hat{z}_{i(c_6)}, \ldots, \hat{z}_{i(c_7)}}^{1 \text{ missed}}, \tag{41}$$

$$\overbrace{\hat{z}_{i(c_6+1)}, \ldots, \hat{z}_{i(c_7)}}^{t_o \text{ over}}, \ldots, \overbrace{\hat{z}_{i(c_8)}}^{t_o-1 \text{ over}}, \ldots, \overbrace{\ldots}^{\ldots}, \ldots, \overbrace{\hat{z}_{i(c_9)}, \ldots, \hat{z}_{i(c_{10})}}^{1 \text{ over}}, \overbrace{\ldots, \hat{z}_{i(k)}}^{\text{perfect}}$$

Then, we compute the metric score $\mathcal{M}(\{y_i\}, \{f_{\text{random}}(y_i)\})$ for each candidate group, and check

if the monotonicity is followed, i.e.

$$
\mathcal{M}(y_i, W_i^{t_w}) < \mathcal{M}(y_i, W_i^{t_w - 1}) < \quad \ldots < W_i^1)
$$
$$
< \mathcal{M}(y_i, M_i^{t_m}) \quad < \mathcal{M}(y_i, M_i^{t_m - 1}) < \ldots < M_i^1) \tag{42}
$$
$$
< \mathcal{M}(y_i, O_i^{t_o}) \quad < \mathcal{M}(y_i, O_i^{t_o - 1}) < \ldots < \mathcal{M}(y_i, O_i^1) < \mathcal{M}(y_i, P)
$$

Where $W_i^t = \{\hat{z}_{i(\gamma)} | \hat{z}_{i(\gamma)}$ has t wrong diagnosis$\}$ (and similarly for $O$, $M$). Then we repeat this with several ($\rho$) samples (x,y) from the dataset to estimate the probability ($\tau$) that metric $\mathcal{M}$ follows clinically applicable monotonic order.

### 5.2 Prevalence invariance

Promising computational techniques today are heavily reliant on data volume, and classification in long-tailed datasets still pose a significant challenge. Thus, if a diagnostic system performs well on one class and poorly on another, but, it just so happens, that very few instances from the poor performance class is encountered, some metrics might fail to accurately assess this weakness (see Table 5). Since imbalanced datasets are almost the norm in diagnostics, it is paramount that metrics pick up on these potential blind spots.

To test for this property, we select two classes from the dataset $a_M$, $a_m$ which are the most and least frequently occurring classes respectively. Then we create a subset $\mathcal{D}_\alpha \subset \mathcal{D}$ of such that

$$
\mathcal{D}_\alpha = \{\hat{z}_i | P(a_M \in \hat{z}_i) = \alpha, \ \text{and}, \ P(a_m \in \hat{z}_i) = 1 - \alpha \, \forall i \in \{1, \ldots, l\}\} \tag{43}
$$

Thus the dataset contains roughly $l\alpha$ instances of class $a_M$ and $l(1 - \alpha)$ instances of $a_m$.

Then we generate predictions $g_{\text{random}}(\mathcal{D}_\alpha)$ based on $\mathcal{D}_\alpha$ just as in the previous section with sensitivity $p_M$, $p_m$ for $a_M$, $a_m$ respectively (and specificity $q$). The quantity we are interested in calculating is the standard deviation of the metric, $\sigma(\mathcal{M})$, which will measure the amount of variation it has when subjected to variations in the dataset. This is given as

$$
\sigma(\mathcal{M}) = \left( \underset{\alpha \sim U(0,1)}{\mathbb{E}} \left[ \mathcal{M}(\mathcal{D}_\alpha, g(\mathcal{D}_\alpha))^2 \right] - \underset{\alpha \sim U(0,1)}{\mathbb{E}} \left[ \mathcal{M}(\mathcal{D}_\alpha, g(\mathcal{D}_\alpha)) \right]^2 \right)^{\frac{1}{2}} \tag{44}
$$

We estimate this quantity with a monte-carlo simulation, by drawing $\eta$ samples from $U(0, 1)$

## 6 Results

In this section, we will analyze outcomes from MedTric and other relevant contenders in the two experiments over three datasets as described in the preceding section.

We use $\rho = 100$ samples from the datasets to probe each metric for monotonicity with 4 pairs of (p,q) and repeat the experiment $n = 10$ times to gather statistics. Unsurprisingly, only MedTric obeys monotonicity 100% of the time (see Fig 3). Note that subset accuracy and hamming loss never obeys expected clinical ordering, thus making them least suited for evaluation of diagnostic systems. Summary of the results can be found in S5 Table in S3 Appendix.

For measuring dispersion, we set $p_M = 0.9$ ("good" performance for abundant class), and $p_m = 0.5$ ("poor" performance for rare class). $\eta = 50$ samples were drawn for $\alpha \sim U(0, 1)$, and for each $\alpha$, $l = 100$ samples for $a_M$, $a_m$ were drawn to create $\mathcal{D}_\alpha$. The experiment was repeated $n = 10$ times each, for $q = 99\%$, 95%, and over all three datasets (see Fig 4).

We consistently observe that our metric has the least dispersion, and is therefore most likely to capture weaknesses of diagnostic systems which would otherwise be obfuscated by rarity. Summary of the results can be found in S6 Table in S3 Appendix.

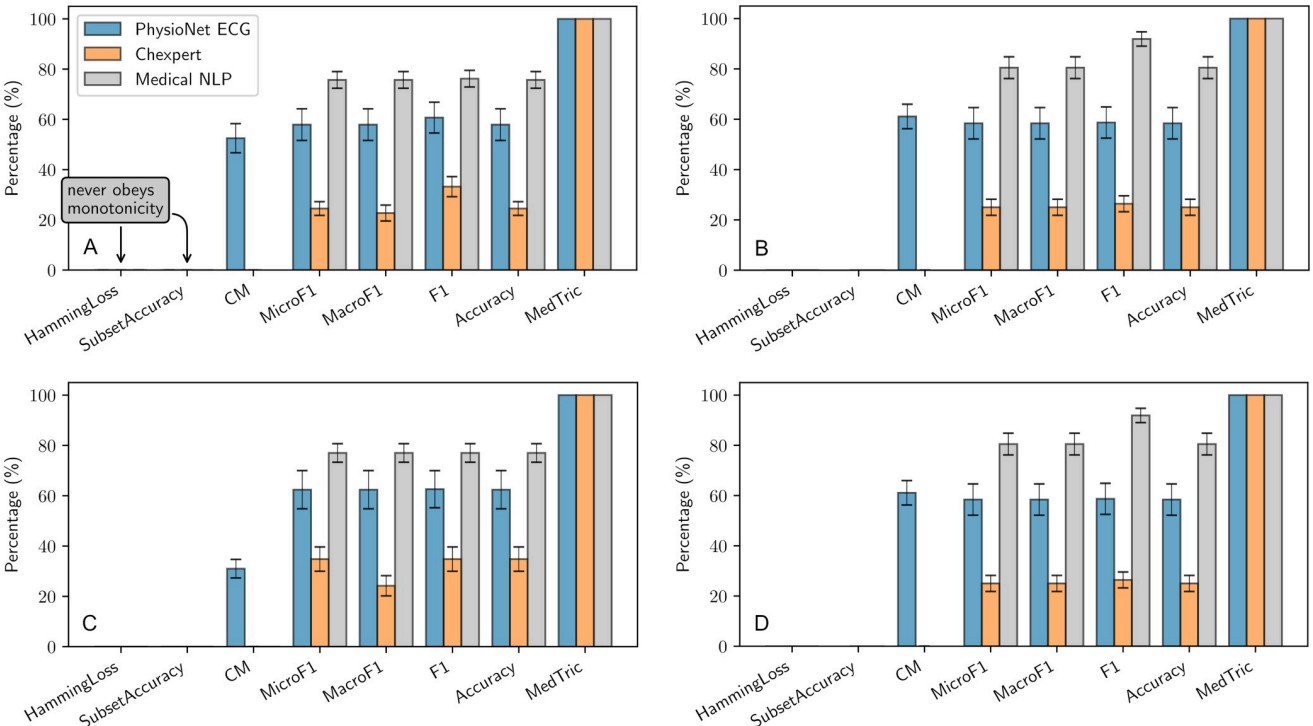

**Fig 3. MedTric is the only metric maintaining clinically applicable order 100% of the time.** The X axis displays the metric under evaluation, and the Y axis shows the percentage of times monotonicity is followed by a particular metric. The experiment is carried out with 4 sensitivity and specificity settings *A*—(80%, 95%), *B*—(80%, 90%), *C*—(60%, 95%), *D*—(60%, 90%) over the three datasets. Hamming loss, and subset accuracy never follows monotonicity. CM was only computed on PhysioNet dataset.

## 7 Conclusions

Current metrics for multi-label computational diagnostics fall short when it comes to capturing the complexities of clinical practice. In particular, we have demonstrated that the commonly used metrics for the bipartition task, does not handle the risks associated with missed diagnosis, over diagnosis and wrong diagnosis in a clinically sound manner. Additionally, we have demonstrated that metric outcomes are often befogged by prevalence and not indicative of actual performance. Clinically paramount features, such as relative importance of diagnoses, and penalizing absurd predictions were heretofore missing. Our metric, however, takes care of the key clinical requirements, making it more aligned to clinical practice. It maintains the order relation between different sorts of diagnostic errors in terms of real life cost. It also handles contradictions, clinical significance, and assigns rewards in accordance with diagnostic practice. Higher values of MedTric correlates with a model that performs better in practice, and all computational models tackling the same problem can be compared in a straightforward manner, even if the metric is calculated over datasets of varying diagnostic distributions.

Given that the norm in comparing computational diagnostic systems was to report results on several, often non-overlapping metrics, each with its own perils, this work has been a major milestone.

MedTric also offers some quick heuristic intuitive notions, for example, if three (equally significant) diagnostic conditions are present in the ground truth and one is missed, a score of $\frac{2}{3}$ is

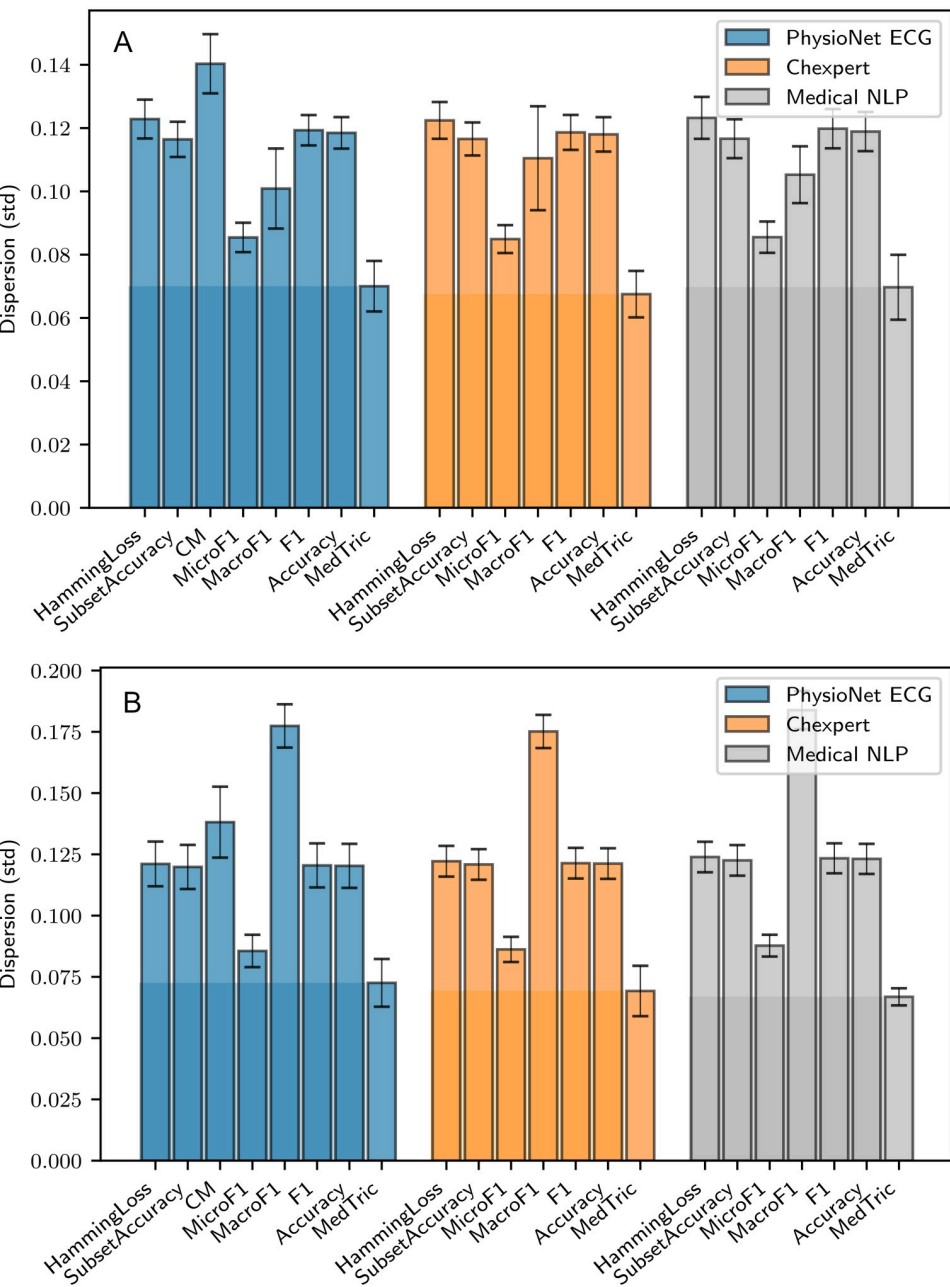

**Fig 4. Dispersion($\sigma$) of various metrics with change in dataset prevalence.** A—$q$ = 95% and B—$q$ = 99%. Metric scores are often dictated by the frequency of occurrence of certain diagnostic conditions in the evaluation dataset, and is not indicative of the actual performance of the computational diagnostic system. High dispersion scores indicate that a metric is likely to obscure weaknesses of diagnostic systems due to relative prevalence of classes. MedTric outperforms other metrics in this regard.

awarded. These features makes usage by humans easier and in addition to dataset independence can serve as a shorthand for the quality of a computational system.

MedTric was designed keeping a clinical setting in mind, however, it can be repurposed for any multi-label classifier evaluation problem, where some domain knowledge can be used to rank different kinds of errors that a computational system can make.

## Supporting information

**S1 Appendix. Binary classification.** This file contains basic definitions and terminology associated with binary classification.
(PDF)

**S2 Appendix. Dataset descriptions and implementation details.** This file contains details of parameters used in our experiments, like values for $w_{ij}$, $C_{ij}$, etc.
(PDF)

**S3 Appendix. Summary of data.** This file contains tables summarizing the experimental data presented in the paper.
(PDF)

## Author Contributions

**Conceptualization:** Soumadeep Saha, Utpal Garain, Arijit Ukil.

**Data curation:** Soumadeep Saha, Sundeep Khandelwal.

**Formal analysis:** Soumadeep Saha.

**Funding acquisition:** Utpal Garain, Arpan Pal.

**Investigation:** Soumadeep Saha, Arijit Ukil, Arpan Pal, Sundeep Khandelwal.

**Methodology:** Soumadeep Saha, Utpal Garain, Sundeep Khandelwal.

**Project administration:** Utpal Garain, Arijit Ukil, Arpan Pal.

**Resources:** Utpal Garain, Arijit Ukil, Arpan Pal.

**Software:** Soumadeep Saha.

**Supervision:** Utpal Garain, Arijit Ukil, Arpan Pal.

**Validation:** Soumadeep Saha, Arijit Ukil, Sundeep Khandelwal.

**Visualization:** Soumadeep Saha.

**Writing – original draft:** Soumadeep Saha.

**Writing – review & editing:** Soumadeep Saha, Utpal Garain, Arijit Ukil.

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
