## [Decision Letter · Decision Letter 0]

21 Oct 2022

PONE-D-22-24889MedTric : A clinically applicable metric for evaluation of multi-label computational diagnostic systemsPLOS ONE

Dear Dr. Saha,

Thank you for submitting your manuscript to PLOS ONE. After careful consideration, we feel that it has merit but does not fully meet PLOS ONE’s publication criteria as it currently stands. Therefore, we invite you to submit a revised version of the manuscript that addresses the points raised during the review process.

We look forward to receiving your revised manuscript.

Kind regards,

Mingxia Liu, PhD

Academic Editor

PLOS ONE

Journal Requirements:

"This work is partially supported by the Science 311and Engineering Research Board (SERB), Dept. of Science and Technology (DST), Govt. of India through Grant File No. SPR/2020/000495."

4. We note you have included a table to which you do not refer in the text of your manuscript. Please ensure that you refer to Table 12 in your text; if accepted, production will need this reference to link the reader to the Table.

**Additional Editor Comments:**

Both reviewers raised several major concerns: 1) insufficient evaluation with only one dataset; 2) missing discussion of other important metrics such as AUC; 3) unclear manuscript organization and low figure resolution.

Reviewers' comments:

Reviewer's Responses to Questions

**Comments to the Author**

1. Is the manuscript technically sound, and do the data support the conclusions?

Reviewer #1: Yes

Reviewer #2: Partly

2. Has the statistical analysis been performed appropriately and rigorously? 

Reviewer #1: Yes

Reviewer #2: No

3. Have the authors made all data underlying the findings in their manuscript fully available?

Reviewer #1: Yes

Reviewer #2: Yes

4. Is the manuscript presented in an intelligible fashion and written in standard English?

Reviewer #1: Yes

Reviewer #2: No

5. Review Comments to the Author

Reviewer #1: This paper focuses on the topic of performance evaluation for multi-label diagnostics. Based on the drawbacks of several widely-used metrics which are borrowed from the machine learning community, the authors propose a new metric, i.e., MedTric, which takes into account several factors that are of clinical importance. Effectiveness is verified through the analysis on examples.

Strengths

[1] The topic of evaluation metrics for medical diagnosis is very interesting which has been overlooked by many research work. Additionally, it has clinical value.

[2] The authors have conducted detailed analysis about the drawbacks of the existing metrics which are directly borrowed from the machine learning community. The proposed metric is based on the characteristics of medical diagnosis system.

[3] The authors have used real datasets to test and verify the effectiveness of the proposed new metric.

Weaknesses

[1] The overall structure of the paper is not quite clear. An “experiment” section is suggested to be highlighted with detailed experimental settings.

[2] The resolutions of the figures are relatively low. The authors are suggested to improve the quality of these figures.

[3] For the analysis part, the discussion of some important metrics are missing. For example, Area under the ROC Curve (AUC) is an important metric for evaluating machine learning-based diagnosis system. The authors are suggested to add some discussions about such metrics.

[4] To verify the advantage of the proposed metric, more datasets should be used for analysis. Since a general-purpose metric can have extensive impact on related research, only using one dataset is not sufficient to verify its utility in real-world medical practice.

[5] In machine learning community, there are some branches, such as cost-sensitive learning which is able to treat different classification results differently. The authors are suggested to add some introductions and discussions about it.

Reviewer #2: The authors proposed a new evaluation index in this paper.

However, since only one public data set is used, the experiment is limited, and clear statistical verification of the experimental results is required.

Also, based on the graph results in figure 5, the proposed medtric indicator does not appear to be superior to other evaluation indicators. The results in figure 6 seem unnecessary.

Finally, authors need to refine the entire manuscript by correcting it with clear English expressions.

6. PLOS authors have the option to publish the peer review history of their article (what does this mean?). If published, this will include your full peer review and any attached files.

Reviewer #1: No

Reviewer #2: No

---

## [Author Response · Author response to Decision Letter 0]

6 Dec 2022

Comments from academic editor

Comment 1: insufficient evaluation with only one dataset.

Response: We wholeheartedly agree with the assessment, and therefore we have added two more datasets for evaluation from different diagnostic disciplines and modalities. In addition to 12 lead ECG data from PhysioNet CINC 2020/21 (line 354) of the marked up manuscript) we have used data from CheXpert, a large scale chest radiography dataset (line 356) and a multi-label clinical free text classification dataset (line 358). In addition to this we have improved the evaluation of the different metrics in question with new statistical measures (line 372, 388) that illustrates the strengths and weaknesses of the metrics better, and our conclusions have been bolstered with more data (line 395, 403).

Comment 2: missing discussion of other important metrics such as AUC.

Response: Thank you for this suggestion. We agree that inclusion of an AUC based discussion would have added value but this is slightly out of scope due to the following reason. AUC provides an aggregate measure of performance across all possible classification thresholds, and indicates how well calibrated the probability outputs are. Computation of AUC (both ROC and PR) hinges on the fact that the diagnostic system in question outputs a probability associated with each class, which may not be true in general. The inference algorithm may be more complicated and choose a set of diagnoses without treating each class as an independent binary classification task, thus making an AUC computation, disconnected from actual predictions. However, if we consider the inference algorithm as part of the blackbox model, we can measure and contrast different paradigms of computational diagnostic systems on a similar footing, as they all need to produce a diagnosis. We have added this note to the manuscript (line 111,123).

Comment 3: unclear manuscript organization and low figure resolution.

Response: Thank you for this comment. We have tried to organize the manuscript better by adding more sections and subsections and clearly outlining the motivation behind each section (line 91,189, 263, 292, 312, 340, 341). We have also postponed additional details to the Supporting Information section. We separated the design of experiments and results into their own sections (line 340, 394). All figures have been recreated in a higher (600 dpi) resolution.

Responses from Reviewers

Comment 1: Is the manuscript technically sound, and do the data support the conclusions?

Reviewer #1: Yes

Reviewer #2: Partly

Response: Thank you for your responses. We further bolstered the conclusions with evaluation on two more datasets (line 356, 358) and redesigned the experiments (line 340) to provide more insights.

Comment 2: Has the statistical analysis been performed appropriately and rigorously?

Reviewer #1: Yes

Reviewer #2: No

Response: We are extremely grateful for your feedback. We have fortified the statistical analysis in our manuscript by adding two statistical measures to support our conclusions (line 372, 388). We have also repeated the experiment across several datasets multiple times to provide further evidence for our conclusions (line 399, 407, S5 Table, S6 Table).

Comment 3: Have the authors made all data underlying the findings in their manuscript fully available?

Reviewer #1: Yes

Reviewer #2: Yes

Response: Thank you for your feedback.

Comment 4: Is the manuscript presented in an intelligible fashion and written in standard English?

Reviewer #1: Yes

Reviewer #2: No

Response: We are grateful for your feedback and we have tried to make changes for the sake of clarity whenever possible, and carefully revised the manuscript (line 4, 23, 72, 80, 82, 84, 88, 134, 160, 191, 279, 315, 395, 447, etc)

Comments from Reviewer 1

This paper focuses on the topic of performance evaluation for multi-label diagnostics. Based on the drawbacks of several widely-used metrics which are borrowed from the machine learning community, the authors propose a new metric, i.e., MedTric, which takes into account several factors that are of clinical importance. Effectiveness is verified through the analysis on examples.

Strengths

Comment 1: The topic of evaluation metrics for medical diagnosis is very interesting which has been overlooked by many research work. Additionally, it has clinical value.

Comment 2: The authors have conducted detailed analysis about the drawbacks of the existing metrics which are directly borrowed from the machine learning community. The proposed metric is based on the characteristics of medical diagnosis system.

Comment 3: The authors have used real datasets to test and verify the effectiveness of the proposed new metric.

Response: Thanks for pointing out that the topic of evaluation metrics for medical diagnosis has been overlooked by many research studies. Lack of such an effort is the central motivation for our research, which was prompted by our need to evaluate models that are more aligned with domain knowledge. We found that existing metrics are inadequate in that regard. We have also added more datasets for evaluation, and statistical measures to buttress our findings.

Weaknesses

Comment 4: The overall structure of the paper is not quite clear. An “experiment” section is suggested to be highlighted with detailed experimental settings.

Response: This comment proved instrumental in improving our manuscript, and we are extremely grateful for your insight. We have tried to organize the manuscript better by adding more sections and subsections and clearly outlining the motivation behind each section (line 91,189, 263, 292, 312, 340, 341). We have also postponed additional details to the Supporting Information section. As per your suggestion, we separated the design of experiments into its own section (line 340) and highlighted experimental settings (line 362, 360, 364, 371, 384, 387, 391, 398, 404, etc)

Comment 5: The resolutions of the figures are relatively low. The authors are suggested to improve the quality of these figures.

Response: We agree with your assessment and have recreated every image at a higher resolution (600 dpi).

Comment 6: For the analysis part, the discussion of some important metrics are missing. For example, Area under the ROC Curve (AUC) is an important metric for evaluating machine learning-based diagnosis systems. The authors are suggested to add some discussions about such metrics.

Response: Thank you for this suggestion, and although we agree with your assessment, in a general discussion of metrics for computational diagnostic systems, this is slightly out of scope. AUC provides an aggregate measure of performance across all possible classification thresholds, and indicates how well calibrated the probability outputs are. Computation of AUC (both ROC and PR) hinges on the fact that the diagnostic system in question outputs a probability associated with each class, which may not be true in general. The AUC metrics are linked to the threshold tuning aspect of machine learning based systems, and is in general not extendable to every kind of model. For example, consider an inference algorithm which chooses a set of diagnoses without treating each class as an independent binary classification task, thus making an AUC computation, disconnected from actual predictions.

However, if we consider the inference algorithm as part of the blackbox model, we can measure and contrast different paradigms of computational diagnostic systems on a similar footing, as they all need to produce a diagnosis. We have added this note to the manuscript (line 111,123).

Comment 7: To verify the advantage of the proposed metric, more datasets should be used for analysis. Since a general-purpose metric can have extensive impact on related research, only using one dataset is not sufficient to verify its utility in real-world medical practice.

Response: We are grateful for your comment, and we have added two more datasets to validate the advantages of the proposed metric, and also highlight issues with existing ones; CheXpert, a large scale chest radiography dataset (line 356) and a multi-label clinical free text classification dataset (line 358). Additionally, we have introduced estimates for quantities like probability of monotonicity violations (line 372) and dispersion with dataset variation (line 388) which quantitatively demonstrates our claims.

Comment 8: In machine learning community, there are some branches, such as cost-sensitive learning which is able to treat different classification results differently. The authors are suggested to add some introductions and discussions about it.

Response: Thank you for pointing this out. Our proposed metric can be thought of as an extension of the cost sensitive learning setup to the clinical setting, and includes a cost term for every possible subset of diagnoses. We have added a few paragraphs highlighting how they fit together (line 76, 323).

Comments from Reviewer 2

The authors proposed a new evaluation index in this paper.

Comment 1: However, since only one public data set is used, the experiment is limited, and clear statistical verification of the experimental results is required.

Response: We agree with your assessment and are grateful for it. We have added two more datasets for evaluation from different diagnostic disciplines and modalities. In addition to 12 lead ECG data from PhysioNet CINC 2020/21 (line 354) we have used data from CheXpert, a large scale chest radiography dataset (line 356) and a multi-label clinical free text classification dataset (line 358). In addition to this we have improved the evaluation of the different metrics in question with new statistical measures like probability of maintaining monotonicity (line 372) and dispersion with change in class frequencies (line 384) that illustrates the strengths and weaknesses of the metrics better, and our conclusions have been bolstered with more data (line 399, 407, S5 Table, S6 Table).

Comment 2: Also, based on the graph results in figure 5, the proposed medtric indicator does not appear to be superior to other evaluation indicators.

Response: Figure 5 was meant to study the variation of the value of a metric as a function of prevalence of a certain class. We outlined that the desirable behavior was minimum variance, or a metric that is as close to constant as possible. Based on our experiments it was clear that MedTric had the least variance, however the graph was a little confusing. Hamming loss stands out from the pack of metrics in the fact that it is the only one with a 1/(number of classes) factor, thus making the scores not in the same scale as all other metrics, however if you account for that fact, it indeed has a large variance. Thus, we have removed figure 5 from the manuscript and devised an experiment to directly estimate the value of the dispersion (line 388) and plotted the same for all metrics and all datasets in question (Fig. 4 in the revised manuscript). Now, it is clear that our proposed metric has the least dispersion, and therefore is most resilient to artifacts arising from prevalence. Thank you for your insightful comments, we believe the change has added to the clarity of the paper.

Comment 3:The results in figure 6 seem unnecessary.

Response: We agree with your assessment that Fig. 6 was unnecessary, and have thus removed it from the manuscript, and just appropriately cited the source.

Comment 4: Finally, authors need to refine the entire manuscript by correcting it with clear English expressions.

Response: Thank you for your feedback. We have tried to make the manuscript clearer whenever possible, and carefully revised the manuscript (line line 4, 23, 72, 80, 82, 84, 88, 134, 160, 191, 279, 315, 395, 447, etc). We have also tried to organize the manuscript better by adding more sections and subsections and clearly outlining the motivation behind each section (line 91,189, 263, 292, 312, 340, 341). Additionally, we have postponed additional details to the Supporting Information section and added a section on experimental design (line 340) so the motivation and methods are clearer. I am confident you will find that the revisions have improved clarity significantly.

---

## [Decision Letter · Decision Letter 1]

14 Feb 2023

PONE-D-22-24889R1MedTric : A clinically applicable metric for evaluation of multi-label computational diagnostic systemsPLOS ONE

Dear Dr. Saha,

Thank you for submitting your manuscript to PLOS ONE. After careful consideration, we feel that it has merit but does not fully meet PLOS ONE’s publication criteria as it currently stands. Therefore, we invite you to submit a revised version of the manuscript that addresses the points raised during the review process.

We look forward to receiving your revised manuscript.

Kind regards,

Mingxia Liu, PhD

Academic Editor

PLOS ONE

Additional Editor Comments (if provided):

Several major concerns from the reviewer need to be clarified, such as inconsistent definitions, insufficient proof, and confused mathematical expressions. Another round of major revision is expected.

Reviewers' comments:

Reviewer's Responses to Questions

**Comments to the Author**

1. If the authors have adequately addressed your comments raised in a previous round of review and you feel that this manuscript is now acceptable for publication, you may indicate that here to bypass the “Comments to the Author” section, enter your conflict of interest statement in the “Confidential to Editor” section, and submit your "Accept" recommendation.

Reviewer #1: All comments have been addressed

Reviewer #3: (No Response)

2. Is the manuscript technically sound, and do the data support the conclusions?

Reviewer #1: Yes

Reviewer #3: Partly

3. Has the statistical analysis been performed appropriately and rigorously? 

Reviewer #1: Yes

Reviewer #3: No

4. Have the authors made all data underlying the findings in their manuscript fully available?

Reviewer #1: (No Response)

Reviewer #3: Yes

5. Is the manuscript presented in an intelligible fashion and written in standard English?

Reviewer #1: Yes

Reviewer #3: No

6. Review Comments to the Author

Reviewer #1: (No Response)

Reviewer #3: (For equations, please see my attachment)

This paper proposed a new metric, i.e., MedTric, to evaluate the multi-label prediction system, which takes the factors that of clinical importance into consideration. Some disadvantages of the general used metrics are discussed. In this revision, most comments from the reviewers have been responded.

However, some important issues remain to be addressed.

1. It looks like that the definitions of “Missed Diagnosis” and “Over Diagnosis” are not consistent when they are used to analyze the disadvantages of the existed metrics and when they are used to define the new metric. Specifically, In Section 2.1, Line 111-114, missed diagnosis and over diagnosis are defined as z ^_k⊊y_k and 〖y_k⊊z ^〗_k. However, In Section 4.1, Line 254-255, 〖y_k-z ^〗_k, z ^_k-y_k correspond to missed diagnosis and over diagnosis, respectively. Also, Fig. 2 is not consistent with the definition in Section 2.1.

Since Missed Diagnosis” and “Over Diagnosis” are the basic concept used in this paper to validate the superiority of the proposed metric, MedTric, this difference should be explained really clear. Otherwise, this inconsistency will bring a big confusion to the readers. Actually, using 〖y_k-z ^〗_k and z ^_k-y_k to represent missed diagnosis and over diagnosis does seem right me.

2. The proof of the Claim in Section 4.2 is not enough to support the “Claim”. Since Equation (39) can only lead to the “Claim” based on the definition of “Missed Diagnosis” and “Over Diagnosis” in Section 4.1 not the one in Section 2.1. Please complete the proof.

3. There might be some mistake in Equation 22. Since o_i is over diagnosis and 〖o_i⊊y〗_i, it looks like 〖〖|y〗_i-m〗_i |=〖〖|o〗_i-m〗_i | will not happen. Please make this equation right or explain it.

4. In Section 3.2, which is the object that the “inequality” in the sentence “the inequality doesn’t hold in general” (Line 225 and 233) represents is not clear to me. Actually, in the example shown in Line 226-227, the inequality (25) holds well. Please make this analysis clearer.

5. The mathematical expressions in this paper needs improvement. There are lots of typos of the mathematical symbols. I’ll list some mistakes here, but the modification should not only limit to these.

1) All the x ^_i in Table 3 should be z ^_i.

2) In Line 279-280, z_i should be z ^_i.

3) g should be g_θ in Eqn. (1).

4) j in Eqn. (23) (26) (28) should be i.

5) Line 244, it’s better to replace z_1,z_2 with z ^_1, z ^_2.

6) j in Table 3 should be i.

7) The usage of index “i”, “j”, “k” should be regulated. The misuse led to lots of confusion.

6. The English expression still need to be improved.

7. PLOS authors have the option to publish the peer review history of their article (what does this mean?). If published, this will include your full peer review and any attached files.

Reviewer #1: No

Reviewer #3: No

---

## [Author Response · Author response to Decision Letter 1]

22 Feb 2023

Comments from Academic Editor

Several major concerns from the reviewer need to be clarified, such as inconsistent definitions, ...

We are extremely sorry for this confusion, and are thankful to you for bringing this to our attention.

The definition for missed diagnosis and over-diagnosis (line 109-114) in Section 2.1 ($\\hat{z}_k \\subsetneq y_k$ and $y_k \\subsetneq \\hat{z}_k$) is the intended definition.

The usage in Section 4.1 (line 257) was colloquial, and we used ``Missed diagnosis'' to mean the diagnoses that were missed (and similarly for over-diagnosis).

We have removed this colloquial usage of ``missed diagnosis'' and ``over-diagnosis'' (line 257) and have started referring to the sets $y_k - \\hat{z}_k, \\hat{z}_k- y_k$ as missed predictions and extra predictions to avoid this confusion.

For the sake of thoroughness we have added definitions for the same (line 115) and have also amended Figure 2 to reflect this.

All our experimental results follow the intended definition, and only one section (Section 4) contained the alternate usage, which has been amended.

We hope this will clear up the inconsistency.

... insufficient proof,...

Thank you for your suggestion and we agree with your assessment. We believe this issue stems from the conflicting usage of missed and over-diagnosis. We have changed the terminology in this revision and our claim has been reworded to state ``MedTric always penalizes missed predictions more severely than extra predictions'' (line 281). The proof which intended to show that the design of our metric is in line with clinical expectations is now complete (line 284).

 Since this alone doesn't demonstrate metric follows required clinical ordering (line 285), we have also added some discussion on how MedTric maintains clinical severity based ordering using the intended definition (Section 2.1, line 109-114), and showed that within certain conditions MedTric guarantees that this ordering holds, and also pointed out the extremes where it would fail. We hope that this additional analysis will further substantiate our claims. The experimental results we reported consists of a fair head-to-head comparison of all metrics on three datasets, and they clearly show that MedTric always maintains the required order even when no other metric does. 

... and confused mathematical expressions. Another round of major revision is expected.

Thank you for your suggestion, we have incorporated all the changes suggested by the reviewers and revised the manuscript to ensure that the mathematical notations are consistent throughout the manuscript.

As reviewer \\#3 pointed out, Equation 22 (line 226) contained a typo, and we have since corrected it.

Hamming Loss is symmetric in missed and extra predictions and scores them equivalently, hence the correct equation would have been $|y_i-m_i|=|o_i-y_i|$. 

Since the usage of dummy indices were confusing we have reserved the index ``i'' to refer to dataset instances and the indices ``j'',``k'' to refer to diagnostic conditions throughout the paper so further confusion can be alleviated.

We hope that the changes made in this revision add clarity and consistency and the added discussions bolster our claims.

Reviewer's Responses to Questions

1. If the authors have adequately addressed your comments raised in a previous round of review and you feel that this manuscript is now acceptable for publication, you may indicate that here to bypass the ``Comments to the Author'' section, enter your conflict of interest statement in the ``Confidential to Editor'' section, and submit your ``Accept'' recommendation.

Reviewer \\#1: All comments have been addressed

Reviewer \\#3: (No Response)

Thank you for your insightful comments in the first round of reviews, your input has been invaluable in improving the manuscript.

2. Is the manuscript technically sound, and do the data support the conclusions?

Reviewer \\#1: Yes

Reviewer \\#3: Partly

Thank you for your responses. Hopefully we have been able to address the confusion regarding the definitions, fix the proofs and the mathematical notations. The added discussions should further highlight the strengths of our metric in a clinical context.

3. Has the statistical analysis been performed appropriately and rigorously?

Reviewer \\#1: Yes

Reviewer \\#3: No

Thank you for your responses. We are hopeful that the changes made in this revision that adds clarity to the definitions and analysis will further fortify our conclusions.

4. Have the authors made all data underlying the findings in their manuscript fully available?

Reviewer \\#1: (No Response)

Reviewer \\#3: Yes

 Thank you for your feedback.

5. Is the manuscript presented in an intelligible fashion and written in standard English?

Reviewer \\#1: Yes

Reviewer \\#3: No

 We are grateful for your feedback and we have tried to make changes for the sake of clarity whenever possible, and carefully revised the manuscript. 

Comments from Reviewer \\#3

This paper proposed a new metric, i.e., MedTric, to evaluate the multi-label prediction system, which takes the factors that of clinical importance into consideration. Some disadvantages of the general used metrics are discussed. In this revision, most comments from the reviewers have been responded. However, some important issues remain to be addressed.

1. It looks like that the definitions of ``Missed Diagnosis'' and ``Over Diagnosis'' are not consistent when they are used to analyze the disadvantages of the existed metrics and when they are used to define the new metric. Specifically, In Section 2.1, Line 111-114, missed diagnosis and over diagnosis are defined as $\\hat{z}_k \\subsetneq y_k$ and $y_k \\subsetneq \\hat{z}_k$. However, In Section 4.1, Line 254-255, $y_k - \\hat{z}_k, \\hat{z}_k- y_k$ correspond to missed diagnosis and over diagnosis, respectively. Also, Fig. 2 is not consistent with the definition in Section 2.1.

Since ``Missed Diagnosis'' and ``Over Diagnosis'' are the basic concept used in this paper to validate the superiority of the proposed metric, MedTric, this difference should be explained really clear. Otherwise, this inconsistency will bring a big confusion to the readers. Actually, using $y_k - \\hat{z}_k$ and $\\hat{z}_k- y_k$ to represent missed diagnosis and over diagnosis does seem right me.

 We are extremely sorry for this confusion, and are thankful to you for bringing this to our attention.

The definition for missed diagnosis and over-diagnosis in Section 2.1 (line 109-114) ($\\hat{z}_k \\subsetneq y_k$ and $y_k \\subsetneq \\hat{z}_k$) is the intended definition.

The usage in Section 4.1 (line 257) was colloquial, and we used ``Missed diagnosis'' to mean the diagnoses that were missed (and similarly for over-diagnosis).

We have removed this colloquial usage of ``missed diagnosis'' and ``over-diagnosis'' and have started referring to the sets $y_k - \\hat{z}_k, \\hat{z}_k- y_k$ as missed predictions and extra predictions (line 257) to avoid this confusion.

For the sake of thoroughness we have added definitions for these terms (line 115) and have amended Figure 2 to reflect this.

We hope this will clear up the confusion.

Although $y_k - \\hat{z}_k, \\hat{z}_k- y_k$ seem intuitive as definitions for missed and over-diagnoses, we are using these terms to talk about the diagnosis as a whole, i.e. $\\hat{z}_i$. For example if a patient has conditions $\\{A,B,C\\}$ and is diagnosed with $\\{A,B\\}$ we call that a missed diagnosis. This is done for two reasons. 

a) It is more obvious what the ordering should be between different diagnoses in a clinical setting by our definition, as by the alternate definition over and missed diagnoses might be present simultaneously. For example if the patient in question receives two diagnoses $\\{A,B,D\\}$ and $\\{A,B\\}$ it is a priori hard to say which is of better quality.

b) It is often not feasible to partition the score up so that we can assign each prediction its own score, for example, accuracy is defined as $\\frac{1}{N} \\sum_i \\frac{\\hat{z}_i \\cap y_i}{\\hat{z}_i \\cup y_i}$, what is the score due to elements in $\\hat{z}_i - y_i$? How do they change if we add an element? To avoid issues like these we felt it was better to talk about the instance as a whole.

2. The proof of the Claim in Section 4.2 is not enough to support the ``Claim''. Since Equation (39) can only lead to the ``Claim'' based on the definition of ``Missed Diagnosis'' and ``Over Diagnosis'' in Section 4.1 not the one in Section 2.1. Please complete the proof.

We wholeheartedly agree with your assessment and we believe this is due to the conflicting usage of missed and over-diagnosis. We have changed the terminology in this revision and $y_k - \\hat{z}_k, \\hat{z}_k- y_k$ are now called missed, and extra predictions respectively. Our claim has been reworded to state ``MedTric always penalizes missed predictions more severely than extra predictions'' (line 281), which was a design goal.

Based on your suggestion we have also added some discussion (line 285-319) on how MedTric maintains clinical severity based ordering using the original definition (Section 2.1, line 109-114), and showed that within certain conditions MedTric guarantees that this ordering holds, and also pointed out the extremes where it would fail. The experimental results we reported consists of a fair head-to-head comparison of all metrics, and they clearly show that MedTric maintains the required order even when no other metric does.

3. There might be some mistake in Equation 22. Since $o_i$ is over diagnosis and $o_i \\subsetneq y_i$, it looks like $|y_i-m_i|=|o_i-m_i|$ will not happen. Please make this equation right or explain it.

Thank you for pointing this out, it was a typo, and we apologise for the confusion. Hamming Loss is symmetric in missed and extra predictions and does scores them equivalently, hence the correct equation would have been $|y_i-m_i|=|o_i-y_i|$. We have corrected the same in the manuscript (line 221) 

4. In Section 3.2, which is the object that the ``inequality'' in the sentence ``the inequality doesn’t hold in general'' (Line 225 and 233) represents is not clear to me. Actually, in the example shown in Line 226-227, the inequality (25) holds well. Please make this analysis clearer.

Sorry about the confusion, we were referring to Eq. 6 in that comment. 

\\begin{equation*}

 \\text{score}_{WD} < \\text{score}_{MD} < \\text{score}_{OD} < \\text{score}_{PD} 

\\end{equation*}

The example you alluded to do in Line 226-227 and the inequality in Eq. 25 holds well, however this behaviour is contrary to expectations in a clinical setting where we expect missed diagnoses to carry harsher penalties than over-diagnosis. To avoid this confusion further we have named the equation (Equation 6) ``Clinical Order'' and we have referred to it by name whenever relevant (line 227, 232, 236, etc)

5. The mathematical expressions in this paper needs improvement. There are lots of typos of the mathematical symbols. I’ll list some mistakes here, but the modification should not only limit to these.

a) All the $x_i$ in Table 3 should be $\\hat{z}_i$.

b) In Line 279-280, $z_i$ should be $\\hat{z}_i$

c) $g$ should be $g_{\\theta}$ in Eqn. (1).

d) $j$ in Eqn. (23) (26) (28) should be $i$.

e) Line 244, it’s better to replace $z_1, z_2$ with $\\hat{z}_1, \\hat{z}_2$.

f) $j$ in Table 3 should be $i$.

g) The usage of index ``i'', ``j'', ``k'' should be regulated. The misuse led to lots of confusion.

Thank you for your suggestion, and we have made all the changes you suggested and revised the manuscript to ensure that the mathematical notation is consistent across the manuscript. We have also reserved the index ``i'' to refer to dataset instances and the indices ``j'',``k'' to refer to diagnostic conditions throughout the paper so further confusion can be alleviated.

6. The English expression still need to be improved.

Thank you for your input, and we have revised the manuscript again in order to improve clarity. We hope you will pardon any transgressions arising due to stylistic or subjective choices.

---

## [Decision Letter · Decision Letter 2]

20 Mar 2023

MedTric : A clinically applicable metric for evaluation of multi-label computational diagnostic systems

PONE-D-22-24889R2

Dear SAHA,

We’re pleased to inform you that your manuscript has been judged scientifically suitable for publication and will be formally accepted for publication once it meets all outstanding technical requirements.

Kind regards,

Mingxia Liu, PhD

Academic Editor

PLOS ONE

Additional Editor Comments (optional):

Reviewers' comments:

Reviewer's Responses to Questions

**Comments to the Author**

1. If the authors have adequately addressed your comments raised in a previous round of review and you feel that this manuscript is now acceptable for publication, you may indicate that here to bypass the “Comments to the Author” section, enter your conflict of interest statement in the “Confidential to Editor” section, and submit your "Accept" recommendation.

Reviewer #1: All comments have been addressed

Reviewer #3: All comments have been addressed

2. Is the manuscript technically sound, and do the data support the conclusions?

Reviewer #1: Yes

Reviewer #3: Yes

3. Has the statistical analysis been performed appropriately and rigorously? 

Reviewer #1: Yes

Reviewer #3: Yes

4. Have the authors made all data underlying the findings in their manuscript fully available?

Reviewer #1: (No Response)

Reviewer #3: Yes

5. Is the manuscript presented in an intelligible fashion and written in standard English?

Reviewer #1: Yes

Reviewer #3: Yes

6. Review Comments to the Author

Reviewer #1: (No Response)

Reviewer #3: The authors have addressed all the concerns that were raised. The modifications made based on the suggestions have significantly improved the quality of the manuscript. I have noticed only a few minor places that may need to be further modified.

1. In Fig.3 and Fig.4, should the name for the algorithm “Chexpert” be changed to “CheXpert”?

2. In Fig.4, if the “std” in the title of yaxis represents the standard deviation, it’s better to change it to “SD” or “Std. Dev.”.

3. In Fig.4 (especially in subfig. B), the key to the colors covers some parts of the bars, it’s better to move it elsewhere or make it smaller.

4. In Line 103, z_{ij} can be changed to z_i{k} (as you used in Eqn. (2)). Since this symbol only appears twice in Line 103 and Eqn. (2), it’s better to make them consistent to avoid confusion.

7. PLOS authors have the option to publish the peer review history of their article (what does this mean?). If published, this will include your full peer review and any attached files.

Reviewer #1: No

Reviewer #3: No

---

## [Editor Report · Acceptance letter]

24 Mar 2023

PONE-D-22-24889R2 

MedTric : A clinically applicable metric for evaluation of multi-label computational diagnostic systems 

Dear Dr. Saha:

I'm pleased to inform you that your manuscript has been deemed suitable for publication in PLOS ONE. Congratulations! Your manuscript is now with our production department. 

Kind regards, 

on behalf of

Dr. Mingxia Liu 

Academic Editor

PLOS ONE